# Ceramic Foam Granulate from Crashed Clinker Pavers

**DOI:** 10.3390/ma19010160

**Published:** 2026-01-02

**Authors:** Alexander Karamanov, Ilian Djobov, Feyzim Hodjaoglu, Lyubomir Aleksandrov, Emilia Karamanova

**Affiliations:** 1Institute of Physical Chemistry, Bulgarian Academy of Sciences, “Acad. Georgi Bonchev” str. bld.11, 1113 Sofia, Bulgaria; i.djobov@ipc.bas.bg (I.D.); feyzim@ipc.bas.bg (F.H.); ekarama@ipc.bas.bg (E.K.); 2Institute of General and Inorganic Chemistry, Bulgarian Academy of Sciences, “Acad. Georgi Bonchev” str. bld.11, 1113 Sofia, Bulgaria; lubomir@svr.igic.bas.bg

**Keywords:** ceramic foam, clinker, reusing, foaming mechanism

## Abstract

The possibility of transforming debris from a ceramic clinker into high quality foam granulate is discussed. The foaming process, which was carried out at temperatures 150–200 °C higher than the production process, was studied by HSM and DTA-TG coupled with MS. Phase and structural transformations were investigated by XRD and SEM, respectively. The results highlight that the foaming mechanism is related to the release of oxygen due to a reduction in Fe^3+^ to Fe^2+^ after the melting of hematite and the dissolution of pseudobrookite present in clinker waste. Granules obtained after 30 min of holding at 1280 °C are impermeable to water and, depending on the cooling applied, have a density between 0.4 and 0.7 g/cm^3^, porosity between 70 and 85 vol %, and compressive strength between 0.7 and 1.1 MPa. These results meet the requirements for high-quality fire-resistance lightweight aggregates.

## 1. Introduction

Good thermal insulation of modern buildings is one of the main criteria to achieve high-energy efficiency and decrease heating costs. Today, probably the most popular products are EPS (Expanded Polystyrene)- and XPS (Extruded Polystyrene)-based panels [1,2], as well as materials from glassy or mineral wool.

However, organic isolation materials are combustible, might adsorb some moisture, and unfortunately do not have good mechanical characteristics. In addition, their lifespan is 30–50 years, so they can be adopted as a temporary solution. Furthermore, their future recycling can be also considered.

Inorganic isolation materials, obviously, are characterized with improved fire resistance and longer durability. In addition, various industrial inorganic wastes might be used as raw materials in their production.

Glass wool insulation is not actually fireproof, but due to its non-flammable glass fibers, it is resistant to relatively high temperatures [3]. The “softening” temperature of glass wools is usually about 600–700 °C, which are typical temperatures for traditional industrial glass compositions [4]. At the same time, mineral wool, also known as rockwool, is characterized with better fire resistance. This non-combustible material can withstand high temperatures, often above 1000 °C, without melting or releasing harmful smoke or gases. This makes it an appreciated component in passive fire protection systems for buildings. The main reason for this performance is the different chemical composition of rockwool, which leads to a partial crystallization of its fibers and thus to increasing its apparent viscosity [5,6,7]. As a result, softening occurs in the melting interval of formed crystal phases [8,9,10].

As alternatives, different natural materials, such as pumice or basaltic tuffs, are also widely used [11,12,13,14,15,16,17]. Due to their porous structure, these lightweight volcanic rocks are suitable for various insulation applications, particularly in construction. They are mainly used in granular form as filler in lightweight concrete blocks, as aggregates in mortars and plasters, or as alternative raw materials for the ceramic industry. Similar to mineral wool, these materials are polycrystalline, which explains their high fire resistance.

It is no coincidence that geological areas with similar rocks are also known for ancient direct rock-cut architecture, famous in Cappadocia and other parts of the world [18,19]. Tuff was also used as a lightweight aggregate added to Roman pozzolan concrete. The most famous example is the Roman Pantheon, where the weight of the massive dome gradually decreases as it rises toward the oculus [20]. This innovative application is key to ensuring the stability of the dome and allows it to stand without reinforcement.

Other advanced isolation materials are glass, ceramic, glass–ceramics, or geopolymer foams, which in many cases also are synthesized by means of various inorganic wastes, as well as using separated appropriated parts of construction and demolition (C&D) wastes.

A significant portion of C&D wastes are different kinds of bricks. If they are not broken their secondary usage is possible. Otherwise, after crushing and eventually milling, they might be used as aggregate in concrete and mortar or as raw materials in new brick production [21,22].

The crashing and milling of traditional missionary bricks, which are quite fragile, is not problematic. However, when facing or clinker bricks, which are characterized by a significantly better degree of sintering and thus have improved mechanical properties, are milled this procedure is not suitable.

Facing and clinker bricks are traditional materials widely used for the exterior cladding of buildings or for paving large public spaces in the early twentieth century. Today, mainly due to their aesthetic rustic appearance, an increase in their manufacture has been observed. A typical modern production line for these materials has a capacity of 10,000–30,000 bricks daily with a scrap rate of about 5–10%.

The present work proposes an alternative approach to use some of these ceramic residues. It reports results for the possibility to transform fired clinker scrap into high quality ceramic foam by a simple thermal treatment at a temperature 150–200 °C higher than the sintering temperature.

## 2. Experimental

In the present study, broken brick pavers from Vitosha Boulevard, which is the main pedestrian zone of the Bulgarian capital Sofia, were used.

Their chemical composition was evaluated with XRF analysis (Zetium Spectrometer–Malvern Panalytical, Malvern Panalytical Ltd., Malvern, UK).

Their thermal behavior was evaluated by non-isothermal treatments at 10 °C min^−1^ with Hot Stage Microscopy (Misura, ESS HSM-1400, Modean, Italy) using little bulk samples and with a DSC-TG apparatus (NETZSCH STA 449 F5 Jupiter, Selb, Germany) equipped with Mass Spectrometer MS (QMS 403 Quadro Aëolos, NETZSCH, Selb, Germany) by means of a powdered sample in a helium (He) atmosphere.

Broken pavers were additionally crashed into fragments weighing about 10–20 g. After that, pairs of these samples were placed on refractory plates and heat-treated in a silite electric furnace (Nabertherm-Bremen, Germany) with a heating rate of 5 °C/min up to different temperatures in the 1250–1310 °C range and held for 30 min. The first sample of these pairs (labeled Q) was rapidly quenched in air, while the second (labeled C) was cooled in the furnace. At these temperatures, the samples bloated, forming ceramic foams that were then cut to observe their structures.

The phase compositions of the “optimal” Q and C samples were compared with that of the parent clinker. In these experiments, XRD patterns were recorded on an Empyrean Powder X-ray diffractometer (Malvern Panalytical, Almelo, The Netherlands) using Cu radiation (λ = 1.5406 Å) at 40 kV and 30 mA and a PIXcel3D detector. Phase identification was carried out using the HighScore Plus program (version 5.3a).

The surface and ruptures of sample C, together with those of the parent clinker, were observed by Scanning Electron Microscopy (JEOL-JSM 6390, Tokyo, Japan) coupled with EDS (INCA) detection after Au metallization.

The porosity, *P*, was evaluated by the following relationship:$$P=100\times \frac{\rho_{as}-\rho_{a}}{\rho_{as}}$$
where *ρ_a_* is apparent density and *ρ_as_* is absolute density. *ρ_a_* was evaluated with a micrometer and balance after precisely cutting samples into a paralepidic shape. *ρ_as_* was measured by gas pycnometer (AccyPy1330, Micromeritic, Norcross, GA, USA) after crashing and milling samples below 75 μm.

To obtain information about the applicability of the material, additional samples suitable for estimation of their compressive strength were prepared. A clinker brick was cut into 3 × 3 × 2 cm samples, which were then heat-treated for 30 min at 1280 °C. As in the previous experiments, some of the samples were rapidly cooled, while others were cooled freely in the furnace. Three test specimens from each series were then cut into rectangular cross-sections with a size of about 4 × 4 cm and about 3 cm in height. In addition, in order to obtain parallel surfaces, the upper and lower planes on which the load was applied were pre-leveled with a fine-grained cement-sand mortar. During the test, the strength of the leveling mortar was much higher than that of the foams. The test was performed using an automatic servo controlled press (UTEST, Ankara, Turkey) at load of 0.05 MPa/s.

## 3. Results and Discussion

Typically, the gas formation in ceramic and glass–ceramic foams is associated with the oxidation of carbon-containing additives (such as graphite, SiC, or organics) or decomposition reactions (mainly due to the addition of Na- or Ca-carbonates) [23,24].

Very interesting alternatives include compositions containing iron oxides, where foaming occurs at a higher temperature and without the addition of bubbling agents in the batches [25,26,27,28]. The synthesis of these glass–ceramic foams is crucially determined by variations in the equilibrium Fe^2+^ ↔ Fe^3+^ within a temperature. It can be assumed that the initial Fe^2+^/Fe^3+^ ratio is close to that of the melting temperature (usually 1350–1450 °C) because it is fixed during melt quenching. Next, during the subsequent thermal treatment, oxidation of the parent glass powders occurs in the glass transition range (i.e., at 600–700 °C), resulting in practically completed Fe^3+^ formation. After that, with increasing temperature the equilibrium Fe^2+^/Fe^3+^ ratio starts to increase, which can lead to a slow partial Fe^3+^ reduction and oxygen release. It can be noted that if the heat treatment occurs in an inert atmosphere no oxidation in the glass transition range and no subsequent foaming occurs.

However, when phase formation takes place during heating and Fe^3+^ ions are incorporated into crystal structures, this reduction happens during the melting of formed crystal phases (usually at about 1100–1200 °C). As a result, if at this moment the apparent viscosity becomes sufficiently low, intensive foaming occurs due to oxygen release into the thus formed pyroplastic matrix. Next, upon cooling, the apparent viscosity again increases, forming a fire-resistant material.

Somewhat similar behavior is also observed in clays containing elevated percentages of iron oxides [29,30]. This phenomenon is different from the formation of “black core” in masonry bricks [31,32], which is the result of an incomplete oxidation of iron oxides to hematite in the bulk of ceramic due to a lack of oxygen during firing or/and when raw materials with higher carbon content are used.

As clinker bricks are manufactured at higher temperatures, technological problems related to their production are more likely to arise as a result of overfiring and deformation than as a result of “black core”.

Ceramic clinker bricks obviously contain mainly high amounts of silica and alumina with some CaO, MgO, and alkali oxides, as well as iron oxides, acting as flux and/or colorant. In fact, the presence of iron oxides in yellow–brown clinker bricks inspired our research with these ceramic residues.

Chemical analysis of used ceramic scrap shows the following results (wt %): SiO_2_—57.4 ± 0.4; TiO_2_—5.2 ± 0.2; Al_2_O_3_—17.4 ± 0.3; Fe_2_O_3_—8.4 ± 0.2; CaO—1.1 ± 0.1; MgO—2.6 ± 0.1; BaO—1.2 ± 0.1; K_2_O—3.6 ± 0.1; and Na_2_O—2.6 ± 0.1. This is a typical composition, the particularity of which is only higher amounts of TiO_2_ and iron oxides.

The XRD pattern of this ceramic is presented in Figure 1a and highlights a relatively high crystallinity, based on residual quartz, together with plagioclase (albite s.s.), pseudobrookite (Fe_2_TiO_5_), and some hematite.

This result coincides with the chemical composition and explains the higher percentage of TiO_2_. Pseudobrookite is a typical yellow–brown pigment for the ceramic sector [33]. It has a high melting temperature, at about 1550 °C, so that, similarly to quartz, it can be considered a filler, increasing high-temperature stability during the production cycle.

This clinker paver is characterized by water absorption of 5.5 ± 0.2, which is typical for these ceramic materials [34]. Values for apparent and absolute densities, which are reported in Table 1, and that of water absorption, correspond to open porosity of about 12% and closed below 2%, which means that the total porosity is more than two times lower than that in common masonry bricks. These characteristics correspond to an intermediate degree of sintering, when the transformation of open into closed porosities starts.

Porosity and XRD results were confirmed by SEM observations.

Figure 2a,b show the surface and fracture of the starting clinker, respectively. At low magnification the surface looks well sintered, but at a higher enlargement (see the inset in Figure 2a) some open pores of 2–4 microns are well distinguished. The fracture demonstrates the relatively high crystallinity and the simultaneous presence of closed spherical pores and residual pores with a non-spherical shape.

The EDS analysis, shown in Figure 2b, elucidates the main crystal phases with the following compositions (wt. %): SiO_2_—98.2, TiO_2_—0.4, Al_2_O_3_—0.8, and Fe_2_O_3_—0.6 for the quartz and SiO_2_—65.0, Al_2_O_3_—21.9, Fe_2_O_3_—0.7, CaO—5.1, Na_2_O—5.6, and K_2_O—1.7 for the plagioclase.

The quartz crystals, some of which were visible even to the naked eye, were between 20 and 60 microns in size, while the albite crystals were between 5 and 20 µm. Both phases evidently were residual, which means that the sintering temperature did surpass 1100–1150 °C.

The HSM curve is shown in Figure 3a. Up to ~1100 °C, volume changes were not observed, so it can be assumed that this was the approximate firing temperature of the clinker. At up to 1150–1160 °C the clinker started re-sintering, leading to 5–6% volume shrinkage, which means that the total porosity decreased about two times. This additional densification indicates that the porosity became mainly closed, which can favor the subsequent foaming.

In fact, with further increases in temperature the sample’s volume started to increase, and at 1280 °C it was near three to four times larger than the initial volume; then the sample went outside the camera’s range.

Silhouettes of the sample at 1100, 1150, 1200, and 1250 °C are shown in Figure 3b. Increasing the temperature up to 1150 °C led to some shrinkage, but no shape change was noted. Some softening was observed at 1200 °C, followed by foaming at 1250 °C.

It can be assumed that after 1150–1200 °C the apparent viscosity rapidly decreased and that some gas formation began. Both processes must be explained by the melting of crystalline phases presented in the parent ceramic.

The DTA-TG analysis, shown in Figure 4, confirmed this assumption. DTA traces show two well-distinguished endo effects, which are explained by the melting of the crystal phases in the clinker. The first effect with an onset temperature of about 1150 °C could be related to the beginning of the formation of the liquids phase at the eutectic temperature [35]. The second endo effect was more intensive and corresponded to the final melting of the crystal phases. It might be presumed that the liquids’ temperature was near the endotherm peak temperature [32], which means that melting could be entirely complete after holding at 1330–1350 °C.

TG data, presented with a traced green line, show practically no variation up to ~1050 °C; then, slow change started, and at 1380 °C the decreasing weight reached about 0.5%, which corresponds to a reduction of 4.4 wt % Fe_2_O_3_ into FeO (i.e., about half of the initial iron oxide in the clinker). This means that the estimated volume of formed oxygen gas phase per a gram sample, corresponding to 1 atmosphere and 1300 °C, was ~20 mL.

The oxygen release as O_2_ was confirmed by MS. These results also are plotted in Figure 4 and show that signals were mainly observed in two intervals: first in the temperature interval of 1050–1150 °C, and the second in 1250–1350 °C (i.e., intervals close to the melting endo effects). No signs of other gases (as CO_2_, CO, SO_2,_ or H_2_O vapors) were indicated during the melting.

After DTA and HSM tests, pairs of larger samples (labeled Q and C) were heat-treated in an electric furnace at 1250, 1270, 1290, and 1310 °C and held for 30 min. Q samples were removed from the furnace at the end of holding and quenched in air, while C samples were free-cooled in the furnace. These specimens were cut, and their photos are presented in Figure 5.

Samples heat-treated at 1250 °C showed somewhat similar structures with evenly distributed closed pores, whereas after treatment at 1270 °C, increasing pore size, leading to better foaming, was observed. In samples that foamed at 1290 °C, pore coalescence was evident, while at 1310 °C a portion of these pores collapsed, which means that the apparent viscosity became too low to support a stable porous structure. These last results are in good agreement with DTA data showing intensive melting of the crystal phases at these temperatures.

A visible contraction of the Q samples, held at 1290 °C and especially 1310 °C, was first observed seconds after their removal from the furnace. This effect might be linked to the fast decrease in gas pressure in the hot pyroplastic matrix during quenching. Subsequently, the apparent viscosity increased sufficiently and the surfaces of samples became “rigid”, fixing the final volume of the foam.

The higher the holding temperature was, the minor the crystallinity and the lower the temperature at which the volume was steadied. Therefore, due to a greater shrinkage during cooling, it is possible that the final porosity of a sample obtained at higher temperature was lower. Similar results have already been reported for a glass–ceramic foam containing iron oxides [28].

The porosity of rapidly cooled samples is visibly higher than that of the corresponding samples cooled in the furnace. Surfaces of the Q series are also smoother and shinier, while C samples formed a wrinkled surface. The reason for this may be the significantly longer time during which C samples remained in a pyroplastic state and the associated larger decrease in gas pressure in the pores.

Finally, a color variation was also observed. Surfaces of the Q series have a dark brown color, while surfaces of Q—light brown indicate the possibility of some reverse FeO into FeO oxidation during slower cooling.

After these tests, final samples treated for 30 min at 1280 °C were prepared.

Two of these specimens are shown in Figure 6, while photographs of their structures are included in Figure 5. XRD patterns of these 1280-Q and 1280-C samples are presented in Figure 1b and Figure 1c respectively.

In order to estimate their porosities, other pairs of samples were carefully cut into parallelepiped form to measure their apparent densities. Next, after milling below 75 micron, their absolute densities were evaluated by gas pycnometry. These results, together with their corresponding porosities, are reported in Table 1 and confirm better foaming at rapid air quenching. Nevertheless, the value for sample C also is quite attractive.

The bloating mechanism was additionally confirmed by XRD results of both foam samples, whose patterns are very similar and demonstrate a significant dissolution of the quartz (the intensity of its peaks decreased more than three times), some diminishing of pseudobrookite, and the entire melting of the plagioclase and hematite. At the same time, traces of the formation of a new spinel phase (close to titaniferrous magnetite) was identified, especially in the slowly cooled sample.

The melting of hematite and the dissolution of a portion of pseudobrookite confirm the proposed mechanism of autocatalytic foaming due to a partial reduction in F_2_O_3_, which occurred during the melting of Fe^3+^ containing crystal phases. The detected formation of titaniferrous magnetite at cooling also supports this hypothesis.

The significant increase in the amorphous phase of these foams explains their lower absolutely density, while the minor density variation of about 0.02 g/cm^3^ between both foams could be mainly related to annealing of the glassy phase in sample C [36]. It is well known that during annealing, glass density increases [37].

The structure of the final 1280-C sample foam was also studied by SEM, and the results for the surface and fracture are presented in Figure 7a and Figure 7b, respectively.

The surface of the foam was non-porous, and at higher magnification some tiny crystals were observed. The fracture confirmed that the porosity was entirely closed, the pores were spherical, and their size reached 2–3 mm.

In order to test an eventual water penetrability, a 1280-Q sample was placed in water. As shown in Figure 8, no visible sinking was observed after 10 days.

Finally, compressive strength results show values of 1.2 ± 0.2 and 0.75 ± 0.25 MPa for the 1280-C and 1280-Q series, respectively. Notwithstanding the primary samples’ preparation, which decreased their mechanical characteristics, these data fit well into Ashby’s density–compressive strength diagram [38] and are close to results for various glass–ceramic foams [39,40,41] and natural products [17,42].

These preliminary results exceed those of popular organic insulations and meet the requirements for high-quality lightweight aggregates. They could be used in construction to produce lightweight concrete or a geopolymer with better thermal and sound insulation and improved seismic characteristics. Typical applications include bridge decks, prefabricated panels, insulation fillers, etc.

## 4. Conclusions

Preliminary results for a low-cost opportunity to transform ceramic clinker scraps into fire-resistant ceramic foam granulate with 70–85 vol % closed porosity are presented and discussed herein.

The bloating mechanism is explained as an autocatalytic process associated with the release of oxygen due to the reduction of Fe^3+^ into Fe^2+^. It occurred during the melting of hematite and partial dissolution of pseudobrookite, which were present in the parent clinker. Optimal foaming was achieved at 1270–1290 °C, which is about 150 °C higher than the production temperature of the ceramic.

It is also elucidated that porosity and appearance depend on the cooling rate. When, after foaming, specimens were rapidly quenched, their porosity was higher and the surface was smoother and shinier; whereas when they were cooled in the furnace porosity decreased and the surface became wrinkled and brownish.

Additional experiments for the evaluation of the optimal heat treatment, including eventually annealing rapidly quenched more porous specimens, are in preparation. It is expected that improved mechanical characteristics will be attained.

## Figures and Tables

**Figure 1 materials-19-00160-f001:**
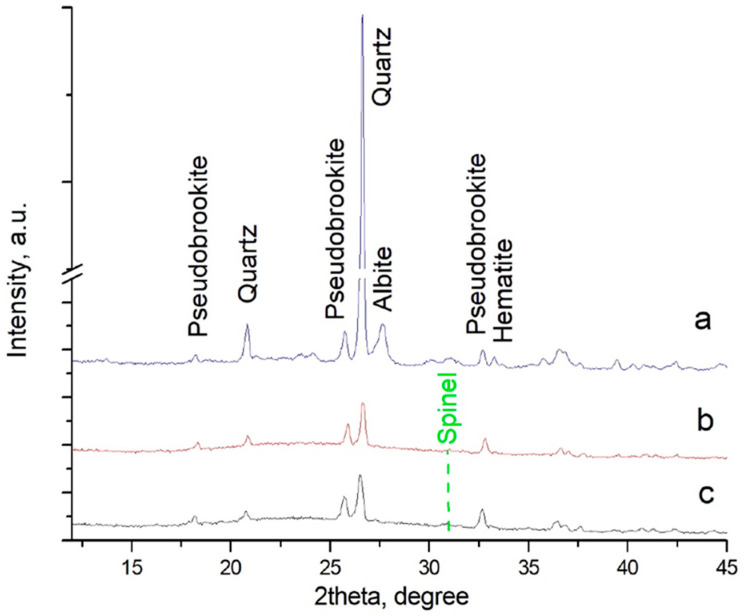
XRD patterns of initial clinker (**a**) and foams 1280-Q (**b**) and 1280-C (**c**).

**Figure 2 materials-19-00160-f002:**
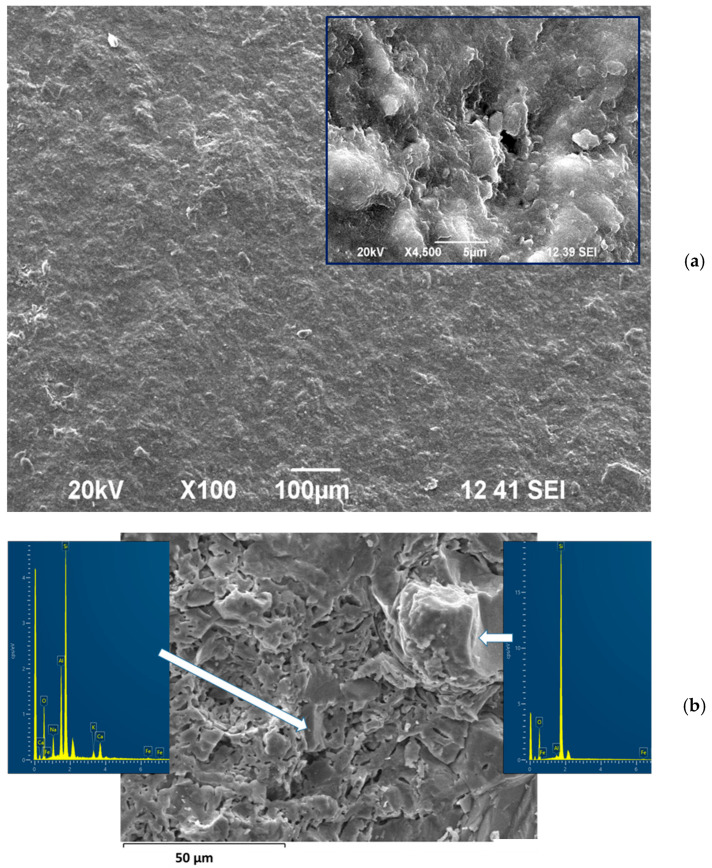
SEM images of surface (**a**) and fracture (**b**) of initial clinker.

**Figure 3 materials-19-00160-f003:**
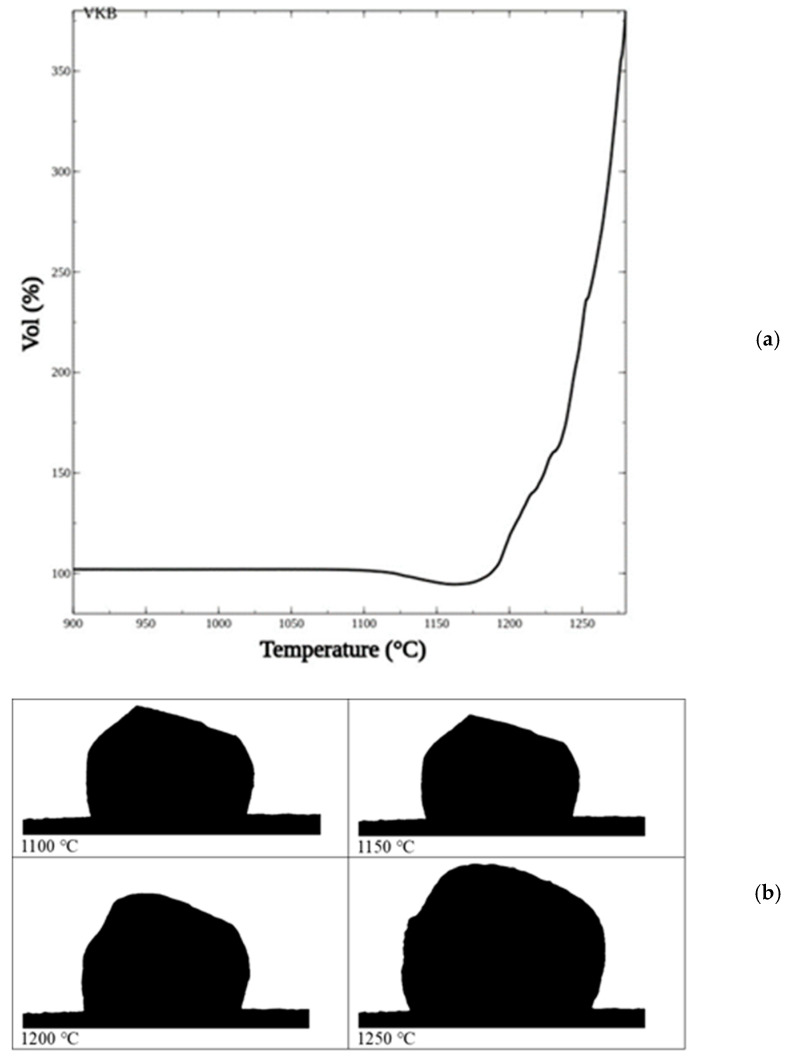
HSM curve (**a**) and sample’s silhouettes at 1100, 1150, 1200, and 1250 °C (**b**).

**Figure 4 materials-19-00160-f004:**
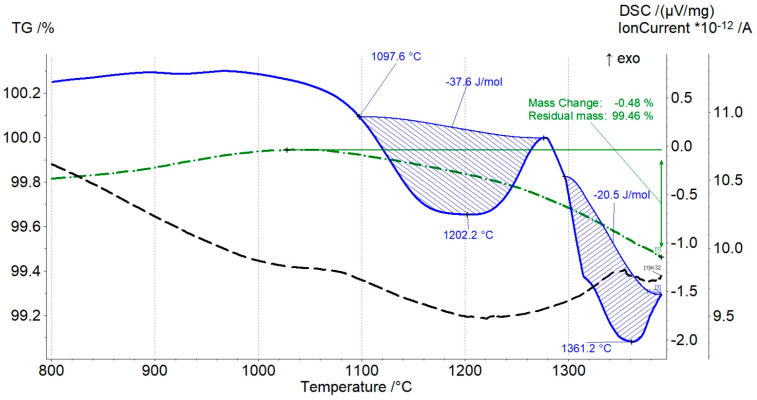
DTA-TG and MS results of milled clinker in He.

**Figure 5 materials-19-00160-f005:**
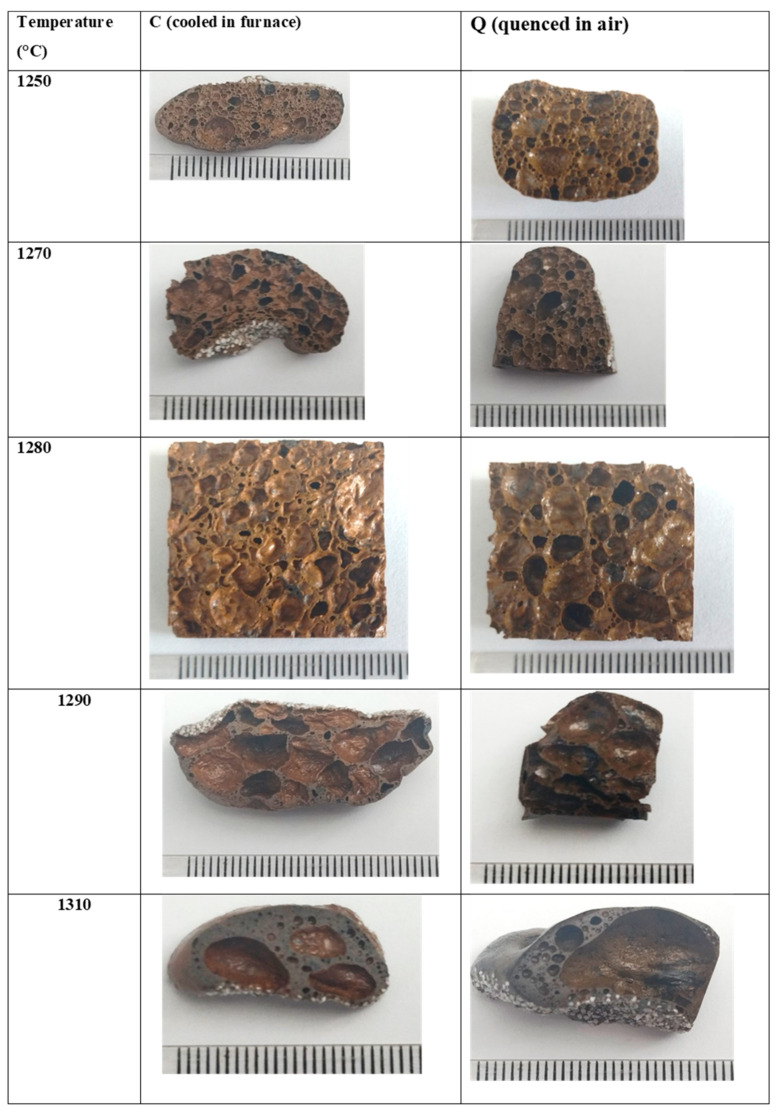
Foam samples, obtained after 30 min holding at different temperatures.

**Figure 6 materials-19-00160-f006:**
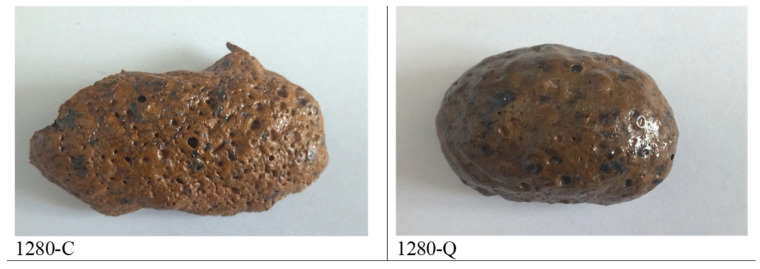
Photos of foams 1280-C and 1280-Q.

**Figure 7 materials-19-00160-f007:**
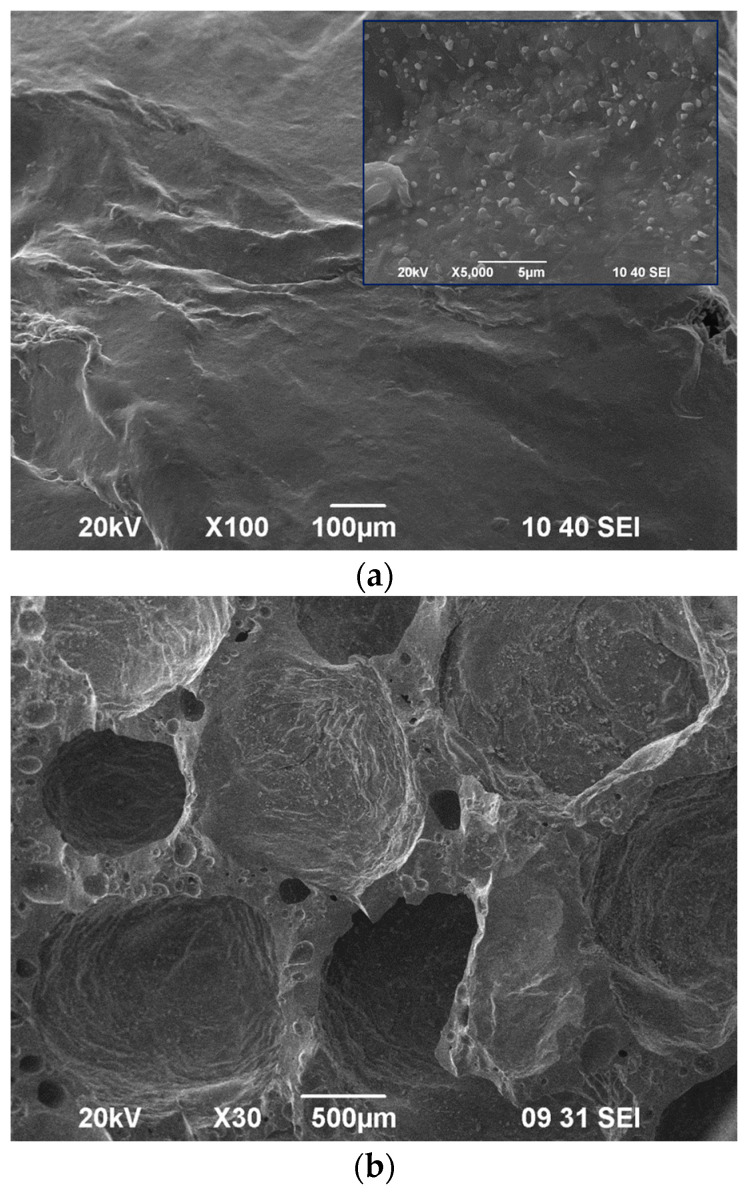
SEM images of surface (**a**) and fracture (**b**) of foam 1280-C.

**Figure 8 materials-19-00160-f008:**
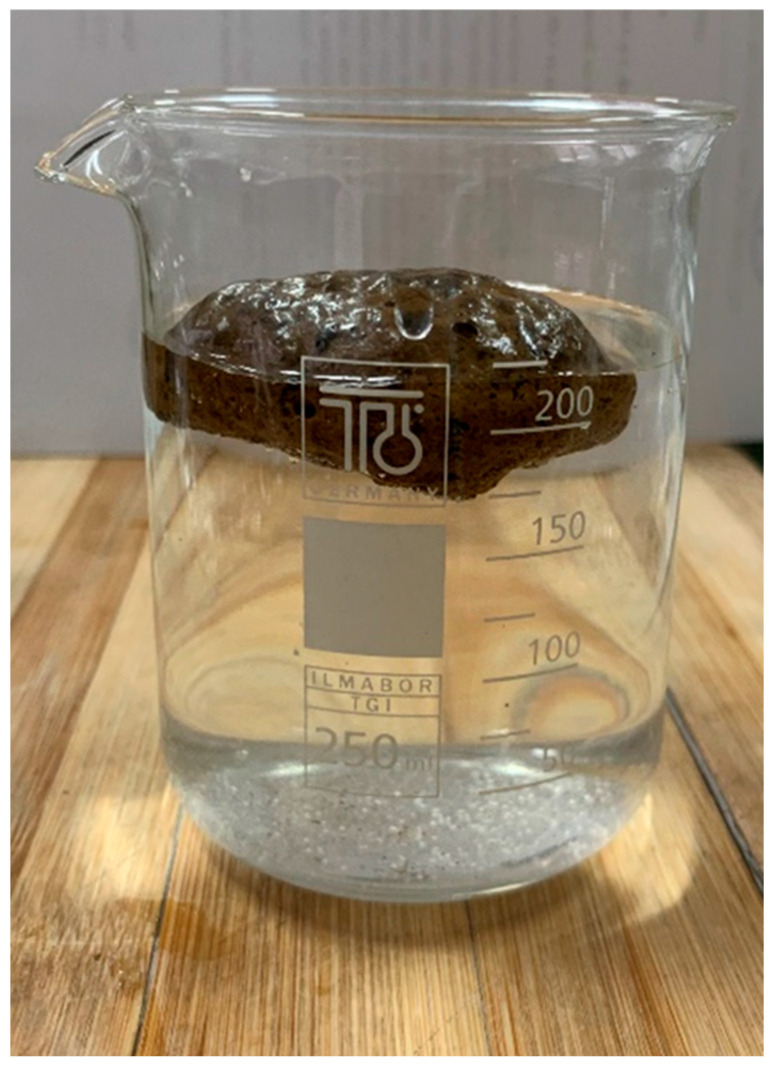
Foam 1280-Q after 10 days in water.

**Table 1 materials-19-00160-t001:** Apparent, ρ_a_, and absolute, ρ_as_, densities and porosity, P.

	ρ_a_ (g/cm^3^)	ρ_as_ (g/cm^3^)	P (vol. %)
Parent clinker	2.33 ± 0.01	2.716 ± 0.002	14.2 ± 0.5
1280-C	0.71 ± 0.03	2.556 ± 0.003	72.2 ± 1.3
1280-Q	0.38 ± 0.03	2.533 ± 0.003	85.0 ± 1.3

## Data Availability

The original contributions presented in this study are included in the article. Further inquiries can be directed to the corresponding author.

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
