# Peer review of "Ceramic Foam Granulate from Crashed Clinker Pavers"

_materials, 2026, doi:10.3390/ma19010160_

Round 1
Reviewer 1 Report
Comments and Suggestions for Authors
The submitted article demonstrates scientific novelty and presents an original and valuable contribution to the field. The experimental procedures are clearly and adequately described. The cited literature is relevant and sufficient. The results are analyzed properly, and the conclusions are well aligned with the research scope and objectives. I recommend minor revision before acceptance.
Please consider the following comments when preparing the final manuscript.
- The abstract should be revised (removing the content indicated in page 1, lines 10–14) and rewritten using short and concise sentences;
- The manuscript could be further improved by presenting the mechanical properties of the developed materials (e.g., compressive strength), as this would provide a more comprehensive evaluation of their performance.;
- How many repeated tests have you done to guarantee the accuracy of the results?
- What potential applications do you envision for these enhanced materials, especially in terms of the industries or construction projects that could benefit from using them?
- Overall, the results are interesting; however, the authors did not provide sufficient discussion or comparison with similar findings reported in the literature. A more thorough reference to relevant studies would strengthen the scientific context and significance of the work;
- Figure 4 quality (Figure 4. DTA-TG and MS results of milled clinker in He). The figure is quite soft and difficult to read at normal size. Please improve contrast, and enlarge scale bars, axes and panel labels so that features and text remain clear after reduction.
These are all minor issues; once addressed, the paper should be suitable for publication.
Author Response
- The abstract should be revised (removing the content indicated in page 1, lines 10–14) and rewritten using short and concise sentences;
The abstract is rewritten:
Тhe possibility to transform debris from a ceramic clinker into high quality foam granulate is discussed. The foaming proses, which carries out at temperatures 150-200 °C higher than the production one, was studded by HSM and DTA-TG coupled with MS. The phase and structural transformations were investigated by XRD and SEM, respectively. The results highlight that the foaming mechanism is related to the release of oxygen due to reduction of Fe3+ to Fe2+ after the melting of hematite and the dissolution of pseudobrookite, present in clinker waste. The granules obtained after 30 minutes of holding at 1280 °C are impermeable to water and, depending on the cooling applied, have a density between 0.4-0.7 g/cm3, porosity between 70 and 85 vol % and compressive strength between 0.7 and 1.1 MPa. These results meet the requirements for high-quality fire-resistance lightweight aggregates.
- The manuscript could be further improved by presenting the mechanical properties of the developed materials (e.g., compressive strength), as this would provide a more comprehensive evaluation of their performance.;
- How many repeated tests have you done to guarantee the accuracy of the results?
New results and comments are added:
To obtain information on the applicability of the material, additional samples suitable for estimation of the compressive strength were prepared. A clinker brick was cut into 3*3*2 cm samples, which were then heat treated for 30 minutes at 1280 °C. As in the previous experiments, some of the samples were rapidly cooled, while others were cooled freely in the furnace. Three test specimens from each series were then cut into rectangular cross-sections with size at about 4*4 cm and at about 3 cm in height. In addition, in order to obtain parallel surfaces, the upper and lower planes on which the load was applied were pre-leveled with a fine-grained cement-sand mortar. During the test, the strength of the leveling mortar was much higher than that of the foams. The test was performed using automatic servo controlled press (UTEST -Turkey) at load of 0.05 МРа/s.
Finally, the compressive strength results show values of 1.2±0.2 and 0.75±0.25 MPa for the 1280-C and 1280-Q series, respectively. Notwithstanding of the primary sample’s preparation, which decreases their mechanical characteristics, these data fit well into Ashby's density-compressive strength diagram [38] and are close to the results for various glass-ceramic foams [39-41] and natural products [17, 42].
These preliminary results exceed those of the popular organic insulations and meet the requirements for high-quality lightweight aggregates. They could be used in construction to produce lightweight concrete or geopolymer with better thermal and sound insulation and improved seismic characteristics. Typical applications include bridge decks, prefabricated panels, insulation fillers, etc.
Tuff was also used as a lightweight aggregate added to Roman pozzolan concrete. The most famous example is the Roman Pantheon, where the weight of the massive dome gradually decreases as it rises toward the oculus [19]. This innovative application is key to ensuring the stability of the dome and allows it to stand without reinforcement.
- What potential applications do you envision for these enhanced materials, especially in terms of the industries or construction projects that could benefit from using them?
- Overall, the results are interesting; however, the authors did not provide sufficient discussion or comparison with similar findings reported in the literature. A more thorough reference to relevant studies would strengthen the scientific context and significance of the work;
New bibliography and new comments, related to these suggestions are added:
- de'Gennaro, R., Cappelletti, P., Cerri, G., de'Gennaro, M., Dondi, M. and Langella, A., 2004. Zeolitic tuffs as raw materials for lightweight aggregates. Applied clay science, 25(1-2), pp.71-81.
- Lancaster, L.C., 2009. Materials and Construction of the Pantheon in Relation to the Developments in Vaulting in Antiquity. In The Pantheon in Rome: Contributions to the Conference Bern. Bern: Bern Studies in the History and Philosophy of Science (pp. 117-125).
- Chinnam, R.K., Francis, A.A., Will, J., Bernardo, E. and Boccaccini, A.R., 2013. Functional glasses and glass-ceramics derived from iron rich waste and combination of industrial residues. Journal of Non-Crystalline Solids, 365, pp.63-74.
38.Michael F. Ashby, Materials Selection in Mechanical Design, Third Edition, Elsevier, Amsterdam, 2025
39.Fernandes, H.R., Tulyaganov, D.U. and Ferreira, J.M.F., 2009. Preparation and characterization of foams from sheet glass and fly ash using carbonates as foaming agents. Ceramics international, 35(1), pp.229-235.
40.Marangoni, M., Secco, M., Parisatto, M., Artioli, G., Bernardo, E., Colombo, P., Altlasi, H., Binmajed, M. and Binhussain, M., 2014. Cellular glass–ceramics from a self foaming mixture of glass and basalt scoria. Journal of non-crystalline solids, 403, pp.38-46.
41.Ge, X., Zhou, M., Fan, C., Zhang, Y. and Zhang, X., 2022. Investigation on strength and failure behavior of ceramic foams prepared from silicoaluminous industrial waste under uniaxial compression. Construction and Building Materials, 317, p.125912.
- Asniar, N., Purwana, Y.M. and Surjandari, N.S., 2019, June. Tuff as rock and soil: Review of the literature on tuff geotechnical, chemical and mineralogical properties around the world and in Indonesia. In AIP Conference Proceedings (Vol. 2114, No. 1, p. 050022). AIP Publishing LLC.
- Figure 4 quality (Figure 4. DTA-TG and MS results of milled clinker in He). The figure is quite soft and difficult to read at normal size. Please improve contrast, and enlarge scale bars, axes and panel labels so that features and text remain clear after reduction.
Fig. 4 is modified.
Reviewer 2 Report
Comments and Suggestions for Authors
In the present manuscript, the authors investigated materials of ceramic foam granulate from crashed clinker pavers. The authors transformed debris from a ceramic clinker by heating it at temperatures between 150-200 °C. The following techniques were employed to study the materials: HSM, DTA-TG coupled with MS, XRD and SEM. The work is interesting, and after revision, can be considered for publication. Points are:
1) The novelty of the work needs to be better highlighted in the introduction section. In addition, more references are needed.
2) Parameters used in the XRF assay need to be added. Also, the Experimental section is short. Please provide more details.
3) Subtopics can be added to the Results and Discussion section to make it easier to read.
4) Relevant information can be highlighted in SEM micrographs. However, EDS spectra are not clearly visible.
5) An analysis of the enthalpies associated with thermal events can be performed using TGA data.
6) Why didn't the authors add chemical composition data from the EDS spectra?
7) What are the prospects for this work? The authors could conclude their work with these prospects.
Author Response
1) The novelty of the work needs to be better highlighted in the introduction section. In addition, more references are needed.
New information and new bibliography is added:
Michael F. Ashby, Materials Selection in Mechanical Design, Third Edition, Elsevier, Amsterdam, 2025
Chinnam, R.K., Francis, A.A., Will, J., Bernardo, E. and Boccaccini, A.R., 2013. Functional glasses and glass-ceramics derived from iron rich waste and combination of industrial residues. Journal of Non-Crystalline Solids, 365, pp.63-74.
Fernandes, H.R., Tulyaganov, D.U. and Ferreira, J.M.F., 2009. Preparation and characterization of foams from sheet glass and fly ash using carbonates as foaming agents. Ceramics international, 35(1), pp.229-235.
Marangoni, M., Secco, M., Parisatto, M., Artioli, G., Bernardo, E., Colombo, P., Altlasi, H., Binmajed, M. and Binhussain, M., 2014. Cellular glass–ceramics from a self foaming mixture of glass and basalt scoria. Journal of non-crystalline solids, 403, pp.38-46.
Ge, X., Zhou, M., Fan, C., Zhang, Y. and Zhang, X., 2022. Investigation on strength and failure behavior of ceramic foams prepared from silicoaluminous industrial waste under uniaxial compression. Construction and Building Materials, 317, p.125912.
de'Gennaro, R., Cappelletti, P., Cerri, G., de'Gennaro, M., Dondi, M. and Langella, A., 2004. Zeolitic tuffs as raw materials for lightweight aggregates. Applied clay science, 25(1-2), pp.71-81.
Asniar, N., Purwana, Y.M. and Surjandari, N.S., 2019, June. Tuff as rock and soil: Review of the literature on tuff geotechnical, chemical and mineralogical properties around the world and in Indonesia. In AIP Conference Proceedings (Vol. 2114, No. 1, p. 050022). AIP Publishing LLC.
Lancaster, L.C., 2009. Materials and Construction of the Pantheon in Relation to the Developments in Vaulting in Antiquity. In The Pantheon in Rome: Contributions to the Conference Bern. Bern: Bern Studies in the History and Philosophy of Science (pp. 117-125).
2) Parameters used in the XRF assay need to be added. Also, the Experimental section is short. Please provide more details.
New results for compressive strength are added.
4) Relevant information can be highlighted in SEM micrographs. However, EDS spectra are not clearly visible.
6) Why didn't the authors add chemical composition data from the EDS spectra?
Fig. 2-b is modified and the EDS results are added in the text.
The EDS analysis, added to Fig. 2-b, elucidate the main crystal phases with the following compositions (wt. %): SiO2 - 98.2, TiO2 - 0.4, Al2O3 - 0.8 and Fe2O3 - 0.6 for the quartz and SiO2 - 65.0,Al2O3 - 21.9, Fe2O3 - 0.7; CaO - 5.1, Na2O - 5.6 and K2O – 1.7 for the plagioclase.
5) An analysis of the enthalpies associated with thermal events can be performed using TGA data.
Fig. 4 is modified.
7) What are the prospects for this work? The authors could conclude their work with these prospects.
Additional information is added in the discussion and the conclusions:
These preliminary results exceed those of the popular organic insulations and meet the requirements for high-quality lightweight aggregates. They could be used in construction to produce lightweight concrete or geopolymer with better thermal and sound insulation and improved seismic characteristics. Typical applications include bridge decks, prefabricated panels, insulation fillers, etc.
Additional experiments for the evaluation of the optimal heat-treatment, including eventually annealing of the rapidly quenched more porous specimens, are in preparation. It is expected that improved mechanical characteristics will be reached.
Round 2
Reviewer 2 Report
Comments and Suggestions for Authors
The authors have made all the requested modifications, therefore I consider the manuscript suitable for publication.